# The Effect of Climatological Variables on Future UAS-Based Atmospheric Profiling in the Lower Atmosphere

**Ariel M. Jacobs** [1,2] , **Tyler M. Bell** [1,2,3,4] , **Brian R. Greene** [1,2,5] **and Phillip B. Chilson** [1,2,5,*]

1   School of Meteorology, University of Oklahoma, 120 David L. Boren Blvd, Ste 5900,
    Norman, OK 73072, USA; ajacobs@ou.edu (A.M.J.); tyler.bell@ou.edu (T.M.B.);
    brian.greene@ou.edu (B.R.G.)
2   Center for Autonomous Sensing and Sampling, University of Oklahoma, 120 David L. Boren Blvd., Ste 4600,
    Norman, OK 73072, USA
3   Cooperative Institute for Mesoscale Meteorological Studies, 120 David L. Boren Blvd., Ste 2100,
    Norman, OK 73072, USA
4   NOAA/OAR National Severe Storms Laboratory, 120 David L. Boren Blvd., Norman, OK 73072, USA
5   Advanced Radar Research Center, University of Oklahoma, 3190 Monitor Ave., Norman, OK 73019, USA
*   Correspondence: chilson@ou.edu

**Abstract:** Vertical profiles of wind, temperature, and moisture are essential to capture the kinematic and thermodynamic structure of the atmospheric boundary layer (ABL). Our goal is to use weather observing unmanned aircraft systems (WxUAS) to perform the vertical profiles by taking measurements while ascending through the ABL and subsequently descending to the Earth's surface. Before establishing routine profiles using a network of WxUAS stations, the climatologies of the flight locations must be studied. This was done using data from the North American Regional Reanalysis (NARR) model. To begin, NARR data accuracy was verified against radiosondes. While the results showed variability in individual profiles, the detailed statistical analyses of the aggregated data suggested that the NARR model is a viable option for the study. Based on these findings, we used NARR data to determine fractions of successful hypothetical flights of vertical profiles across the state of Oklahoma given thresholds of visibility, cloud base level (CBL) height, and wind speed. CBL height is an important parameter because the WxUAS must stay below clouds for the flight restrictions being considered. For the purpose of this study, a hypothetical WxUAS flight is considered successful if the vehicle is able to reach an altitude corresponding to a pressure level of 600 hPa. Our analysis indicated the CBL height parameter hindered the fractions of successful hypothetical flights the most and the wind speed tolerance limited the fractions of successful hypothetical flights most strongly in the winter months. Northwest Oklahoma had the highest fractions of successful hypothetical flights, and the southeastern corner performs the worst in every season except spring, when the northeastern corner performed the worst. Future work will study the potential effect of topology and additional variables, such as amount of rainfall and temperature, on fractions of successful hypothetical flights by region of the state.

**Keywords:** atmospheric boundary layer; meteorology; unmanned aerial systems; local climatology

---

## 1. Introduction

Climate related disasters cost the United States hundreds of billions of dollars a year. Therefore, meteorological measurements are vital for the decision-making process in the economic sectors of energy security, food production, public health and safety, transportation, and water resources [1,2].

These economic factors, along with other considerations, have led to 27 USA states, including Oklahoma, to establish local surface observation networks known as mesonets [3]. These state networks focus on providing real-time, high-quality observations to their citizens and to aid in mesoscale climate and weather monitoring [4]. In Oklahoma, more than 110 stations measure, among other parameters, pressure, wind speed, wind direction, air temperature, solar radiation, soil temperature, and precipitation totals [5]. While surface observations are necessary for monitoring real-time conditions and helping to establish regional climatologies, vertical profiles of wind, temperature, and moisture are essential to capture the kinematics and thermodynamics in the lower atmospheric boundary layer (ABL) [2,6]. The ABL, the layer of the troposphere closest to and directly influenced by the Earth's surface, provides the instability, low-level moisture, low-level wind shear, and lift necessary for convection initiation of thunderstorms [6]. Knowledge of these parameters has the potential to improve weather forecasting during severe storms [7,8].

Despite the importance of vertical profiles in the ABL, the availability of such data are limited, resulting in a "data gap" in the lower atmosphere [2,6]. One possible solution to this problem is to use unmanned aircraft systems (UAS) to perform in-situ, routine vertical profiles. So-called weather observing UAS (WxUAS) would operate in conjunction with mesonet sites to provide the needed ABL measurements from the surface up to heights of 2–3 km above ground level [9]. By extending existing mesonet infrastructure vertically with unattended, automatically flown WxUAS profiles, the 3D mesonet has the potential to improve numerical weather prediction (NWP) models via data assimilation [9]. However, several questions have to be answered before a 3D mesonet can be created. These questions provide a motivation for analyzing the climatological data. The questions include: How high can we fly the WxUAS? How often should we fly them? and What would be the ideal horizontal spacing of the WxUAS stations? Such questions will require further research to address.

Before answering these questions, climatologies of the potential flight locations must be established. This paper outlines a statistical framework in which to analyze the climatological data. This study analyzes data retrieved from the NARR (North American Regional Reanalysis) model to determine how different climatological variables impact the frequency at which and altitude above mean sea level (AMSL) to which the WxUAS would be able to fly in different locations across the state of Oklahoma. The climatological variables were chosen based on the Federal Aviation Administration (FAA) regulations for small UAS and their wind tolerances. Before the model was used, we compared the NARR data with radiosonde data to determine the viability of the model in this study. The paper is structured as follows: in Section 2, we discuss the data and methods used for analysis; in Section 3, we discuss the validation process; in Section 4, we discuss the results of the analysis; in Section 5, we discuss the implications of the results and future work to be done; and, in Section 6, we conclude the paper.

## 2. Data and Methods

### 2.1. Data

#### 2.1.1. NARR Data

The North American Regional Reanalysis (NARR) model is used to generate a regional reanalysis dataset of dozens of variables including winds, temperature, moisture, and visibility [10]. To improve the accuracy of the dataset, the NARR model assimilates observations from pressure data at the surface as well as winds, temperature, and moisture from radiosondes and dropsondes, temperature and winds from aircraft, winds from pilot balloons (pibals), satellite radiance from orbiting satellites, and cloud drift winds from geostationary satellites. Significant advantages of the NARR model include spatial resolution of 32-km, 45 vertical layers, and 3-hourly output [10]. For this study, we used data from 2016, 2017, and 2018 as the period of investigation. Profiles of zonal and meridional winds, geopotential height, temperature, specific humidity, turbulent kinetic energy (TKE), visibility, cloud base pressure, cloud top pressure, cloud base level (CBL) height, cloud top height, and planetary

boundary layer height values were extracted from the NARR data based on the locations of Oklahoma Mesonet sites. To do so, we used the Lambert conformal conic projection to project the latitude and longitude values onto the same grid as the NARR model. We then used the NARR data from the point closest to the geographical location of the mesonet site. Once the data were gathered, they were stored in NetCDF files based on the year the data were collected.

### 2.1.2. Radiosonde Data

To utilize the NARR model to assess the parameters in which the WxUAS would be able to fly, the biases and variability of the model must be understood and addressed. To determine the statistics regarding accuracy in Oklahoma specifically, the NARR values in closest proximity to Norman were compared to the National Weather Service Norman (OUN) radiosonde site. Radiosondes are launched twice a day at 00Z and 12Z and measure pressure, temperature, and moisture variables [11]. The OUN radiosondes are "rawinsondes", which means wind speed and direction are also calculated by the instruments using global positioning system (GPS) technology [11]. Values from OUN Radiosonde data retrieved from the University of Wyoming archive [12] were interpolated to the NARR pressure levels of 100, 125, 150, 175, 200, 225, 250, 275, 300, 350, 400, 450, 500, 550, 600, 650, 700, 725, 750, 775, 800, 825, 850, 875, 900, 925, 950, and 975 hPa. We included the following variables in this process: pressure, height, temperature, dew point, relative humidity, mixing ratio, wind direction, wind speed, potential temperature, equivalent potential temperature, and virtual potential temperature. The interpolated values were stored in NetCDF files in the same format as the NARR data for consistency. The sounding data were collected for 2016, 2017, and 2018.

### 2.2. NARR Validation

Walters et al. compared the wind speed values between the NARR model and 12 radiosonde sites, including OUN. For Norman, Oklahoma, they found the NARR overestimated the winds at the surface for the years 1980, 1990, 2000, and 2010 [13]. In this study, we compared the 2016, 2017, and 2018 data to determine if the winds are still overestimated. Our study also compares the temperature and water vapor mixing ratio values, which has not previously been analyzed in the context of year-round climatologies. To evaluate the accuracy of the NARR data, we performed a detailed statistical analysis of the differences in wind speed, temperature, and water vapor mixing ratio values between the OUN sounding data and the NARR corresponding to the Norman, Oklahoma mesonet site. The Norman site will herein be described as NRMN. We compared the data by meteorological season aggregated over the period of investigation. A large dataset is necessary to account for the different assumptions used in the radiosonde and the NARR model.

### 2.3. Seasonal Climatologies

Oklahoma has dynamic weather, and the wind and cloud conditions change with each season. Therefore, the seasons were analyzed individually. The meteorological seasons are defined as winter (December, January, and February), spring (March, April, and May), summer (June, July, and August), and autumn (September, October, and November). Although the WxUAS considered in this study are not built to withstand extreme weather conditions (i.e., thunderstorms or freezing rain), their utility in observing the ABL leading up to extreme events shows promise in improving numerical weather prediction model forecasts [9,14,15]. For these reasons, it is important to characterize the hypothetical performance of WxUAS in "base state" conditions. Certain limiting parameters were used to determine the fractions of successful hypothetical flights of WxUAS sampling with respect to each season over the period of investigation. For this study, a WxUAS flight is considered to be successful if a hypothetical UAS would be able to reach a given pressure level based on consideration of the atmospheric conditions and FAA regulations. The FAA has set restrictions on the conditions in which UAS are allowed to fly. Most notably, when operating under Visual Line Of Sight (VLOS) restrictions, for example, a surface visibility of three statute miles (4828 m) is required and UAS must stay below 500 feet (152.4 m) of

the CBL height [16]. The same rules apply when piloted aircraft are operated under Visual Flight Rules (VFR) in certain airspace classes. Moreover, we must consider the wind speed tolerance of a particular UAS. The CopterSonde, a UAS developed by the Center for Autonomous Sensing and Sampling (CASS), currently has a wind tolerance of 22 m s$^{-1}$, but this threshold is evolving over time with advancements in the CopterSonde design [17]. We chose the CopterSonde as a representative rotary-wing UAS as it has been extensively tested and experimentally utilized by CASS. Therefore, a wind tolerance of 25 m s$^{-1}$ was used for this study. The study focuses on the extent these parameters affect the fraction of successful hypothetical UAS flights for each meteorological season averaged over the period of investigation. The pressure level considered for flight success was set at 600 hPa (around 4000 m AMSL) because around 4000 m AMSL is the upper bound altitude we expect is necessary to reach to capture the thermodynamic and kinematic structure of the ABL. Each parameter was analyzed individually to determine the seasonal fraction of successful hypothetical flights for each mesonet station only accounting for that variable. This was to determine the amount each parameter restricted the number of successful flights. Next, the parameters were combined to determine the overall fraction of successful hypothetical flights for each season. Once the fractions were calculated for each mesonet site, the data were interpolated across the state of Oklahoma using the Scipy non-uniform griddata function [18]. Table 1 shows the thresholds that result in unsuccessful hypothetical flights.

**Table 1.** The conditions for each parameter that result in unsuccessful hypothetical flights.

| | **Thresholds** |
| --- | --- |
| Visibility | <4828 m |
| CBL Height | CBL Height $-$ 152.4 m < 600 hPa altitude & low + mid cloud cover >12% |
| Wind Speed | >25 m s$^{-1}$ |

The FAA mandates several visibility conditions under VLOS rules. At the surface, visibility determines whether the UAS can takeoff. If the surface visibility is less than 4828 m, a UAS flight is not permitted. Moreover, the FAA requires the UAS to remain at least 152.4 m below the bases of clouds, and the UAS cannot fly in clouds for the flight restrictions being considered. An et al. found that the NARR estimates the cloud base to be lower than the true cloud base in humid climates [19]. To account for this, our study filtered out a NARR product of combined low cloud and medium cloud percentage corresponding to clear skies when determining unsuccessful flights. The National Weather Service defines clear skies as 1/8 or less cloud cover [20]. However, the cloud cover is reported in whole number increments, so 12% was used. If 152.4 m below the cloud base level height corresponded to a pressure greater than 600 hPa and a combined low cloud and medium cloud percentage greater than 12%, the hypothetical flight was considered unsuccessful. The maximum height AMSL value was set to the AMSL height converted from the geopotential height corresponding to the lowest successful pressure level. The geopotential height used was a NARR product we retrieved, which had values corresponding to each pressure level.

The study used a wind tolerance of 25 m s$^{-1}$ as an upper bound estimate. If the wind speeds exceeded 25 m s$^{-1}$ during a vertical profile, the hypothetical flight was considered unsuccessful and the maximum height AMSL value was set to he AMSL height converted from the geopotential height corresponding to the lowest successful pressure level.

*2.4. Selected Sites*

Nine sites around Oklahoma were selected to analyze statistics regarding the maximum height AMSL that could be reached by the UAS. The Oklahoma Mesonet sites were chosen based on their location and regional climate to provide a representative sample of potential UAS flight locations. All nine climate divisions of Oklahoma are represented in the sample [21]. These sites are Fittstown

(FITT), Hobart (HOBA), Idabel (IDAB), Kenton (KENT), Medford (MEDF), Miami (MIAM), Norman (NRMN), Porter (PORT), and Putnam (PUTN) and are shown in Figure 1. Table 2 shows the latitude, longitude, climate divisions, and distances to the nearest NARR grid point of the selected Mesonet sites in Figure 1. For each site, the maximum potential UAS sampling height AMSL statistics were calculated by season given the parameters of wind speed individually, CBL height individually, and combined conditions of wind speed, CBL height, and visibility. For the combined conditions maximum potential UAS sampling height AMSL statistics, the maximum height AMSL was set to the site's elevation if the visibility was below the required threshold. We did not plot visibility separately because visibility determines if a UAS flight is permitted, not how high the UAS can reach after takeoff.

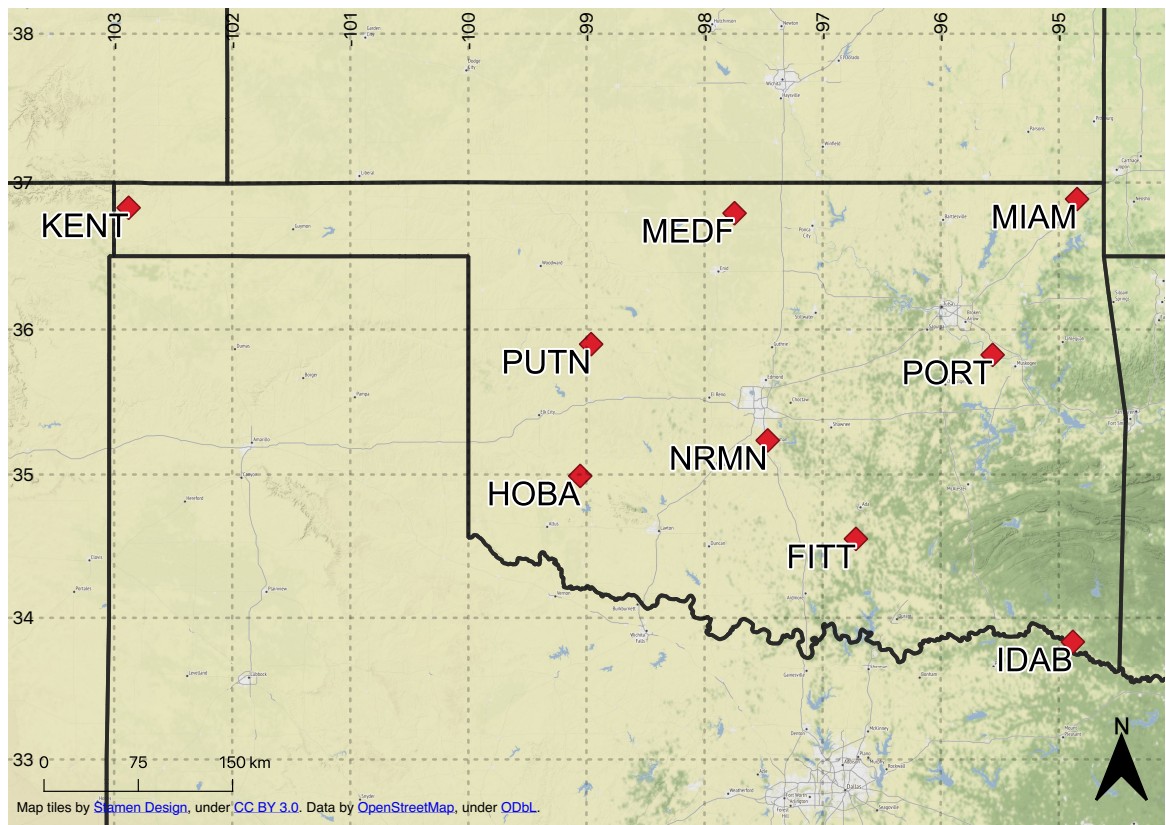

**Figure 1.** Selected Mesonet sites from each of the nine climate divisions of Oklahoma.

**Table 2.** The latitude, longitude, climate divisions, and distances to the nearest NARR grid point of the selected Mesonet sites in Figure 1.

| Site ID | Latitude (°N) | Longitude (°E) | Climate Division | Nearest NARR Grid (km) |
|---------|---------------|----------------|------------------|------------------------|
| FITT | 34.55205 | −96.71779 | South Central | 17.55 |
| HOBA | 34.98971 | −99.05283 | Southwest | 18.57 |
| IDAB | 33.83013 | −94.88030 | Southeast | 5.14 |
| KENT | 36.82937 | −102.8782 | Panhandle | 3.89 |
| MEDF | 36.79242 | −97.74577 | North Central | 14.88 |
| MIAM | 36.88832 | −94.84437 | Northeast | 20.56 |
| NRMN | 35.23611 | −97.46488 | Central | 8.55 |
| PORT | 35.82570 | −95.55976 | Northeast | 14.09 |
| PUTN | 35.89904 | −98.96038 | West Central | 19.10 |

## 3. Validation

For each season, statistical comparisons of the temperature, water vapor mixing ratio, and wind speed were performed using differences between the OUN sounding data and the NARR data.

From these values, we calculated the mean difference, median difference, standard deviation, slope, intercept, and Pearson correlation. Since the NARR values were averaged over a spatial grid and the sounding values were interpolated on to the NARR pressure levels, data outside of the second standard deviation range were considered outliers and were excluded from the results. The variables were also compared at the pressure levels up to 600 hPa. Since wind tolerance was used as a limiting parameter in the seasonal climatology study, we focused on validating the wind speed from the NARR using radiosondes from OUN. For brevity, only two seasons, winter and summer, are presented in the main text. The results are shown in Figure 2. In addition to the wind speeds from spring and autumn, similar figures focusing on temperature and water vapor mixing ratio for all seasons can be found in Figures S1–S10 in the supplemental document.

As shown in Figure 2a, the winter NARR wind speeds had a high Pearson correlation of 0.99 with the sounding wind speeds. The mean difference between the wind speeds was 0.38 m s$^{-1}$. In addition, 9963 out of 10,530 data points were within two standard deviations. The median difference was 0.00 m s$^{-1}$, and the standard deviation was 2.16 m s$^{-1}$ as shown in Figure 2b.

For the summer months, 9365 out of 9902 data points corresponding to NARR wind speeds vs. sounding wind speeds were within two standard deviations. The Pearson correlation was lower than that of the winter plot, but it was still quite high at 0.95, and the mean difference was similar at 0.36 m s$^{-1}$ as shown in Figure 2c. Figure 2d showed the standard deviation was 1.79 m s$^{-1}$, lower than that of winter, and a median difference of 0.00 m s$^{-1}$. These statistics indicated a general strong agreement in wind speed between the NARR and the OUN radiosonde, with exceptions for NARR values that overestimated and underestimated the wind speeds.

The differences between the radiosonde values and the NARR values for temperature, water vapor mixing ratio, and wind speed are shown in Figure 3 as a function of altitude. In the interest of brevity, only two seasons, winter and summer, are presented in the main text. The figures for spring and autumn can be found in Figures S11 and S12. In Figure 3a, the mean winter temperature differences hovered around zero except at 975 hPa where the mean difference is about 1 K. The scatter of the temperature differences generally decreased with pressure. The largest temperature dispersion was at 950 hPa from just under −7.5 K to just over 5 K. The majority of outliers were NARR overestimates of the temperature between 900 and 700 hPa. In Figure 3b, the winter water vapor mixing ratio mean differences were around zero g kg$^{-1}$, and the values were consistently between −2 and 2 g kg$^{-1}$ for all pressure levels between 975 and 600 hPa. There were outliers that were overestimates and some that were underestimates. Figure 3c showed on average, the NARR overestimated the winter wind speeds at 975 hPa. For the remainder of the profile, the interval of differences ranged from a lower bound between −10 and −5 m s$^{-1}$ to an upper bound between 7 and 14 m s$^{-1}$ depending on the pressure level. The means from 950 to 600 hPa show the NARR underestimates the wind speeds at about 1 to 2 m s$^{-1}$.

We ignored the 975 hPa pressure level in the summer plots because there were only three data points, which were not necessarily representative of the entire sample. The summer temperature plot, Figure 3d, showed that the mean temperature difference increased from −1 K up to 0 K with increasing altitude (up to 700 hPa). The scatter of the temperature differences generally stayed between −5 and 2.5 K and the interval shrank as the pressure decreased. The largest range of scatter was at 950 hPa from about −10 K to about 4 K. There are outliers above and below the mean with most of the outliers being NARR overestimates of the temperature. The water vapor mixing ratio means in Figure 3e were around zero g kg$^{-1}$. The range of differences was larger than the range for winter with most values between −4 and 4 g kg$^{-1}$ for the profiles. There are outliers indicating overestimates and underestimates. Figure 3f showed on average, the NARR underestimated the wind speeds by about 1 to 2 m s$^{-1}$. Throughout the profiles, the differences ranged from −13 to 15 m s$^{-1}$ with the majority of differences within −5 to 10 m s$^{-1}$. Most outliers were NARR underestimates of the wind speeds, which will be discussed later on in this section.

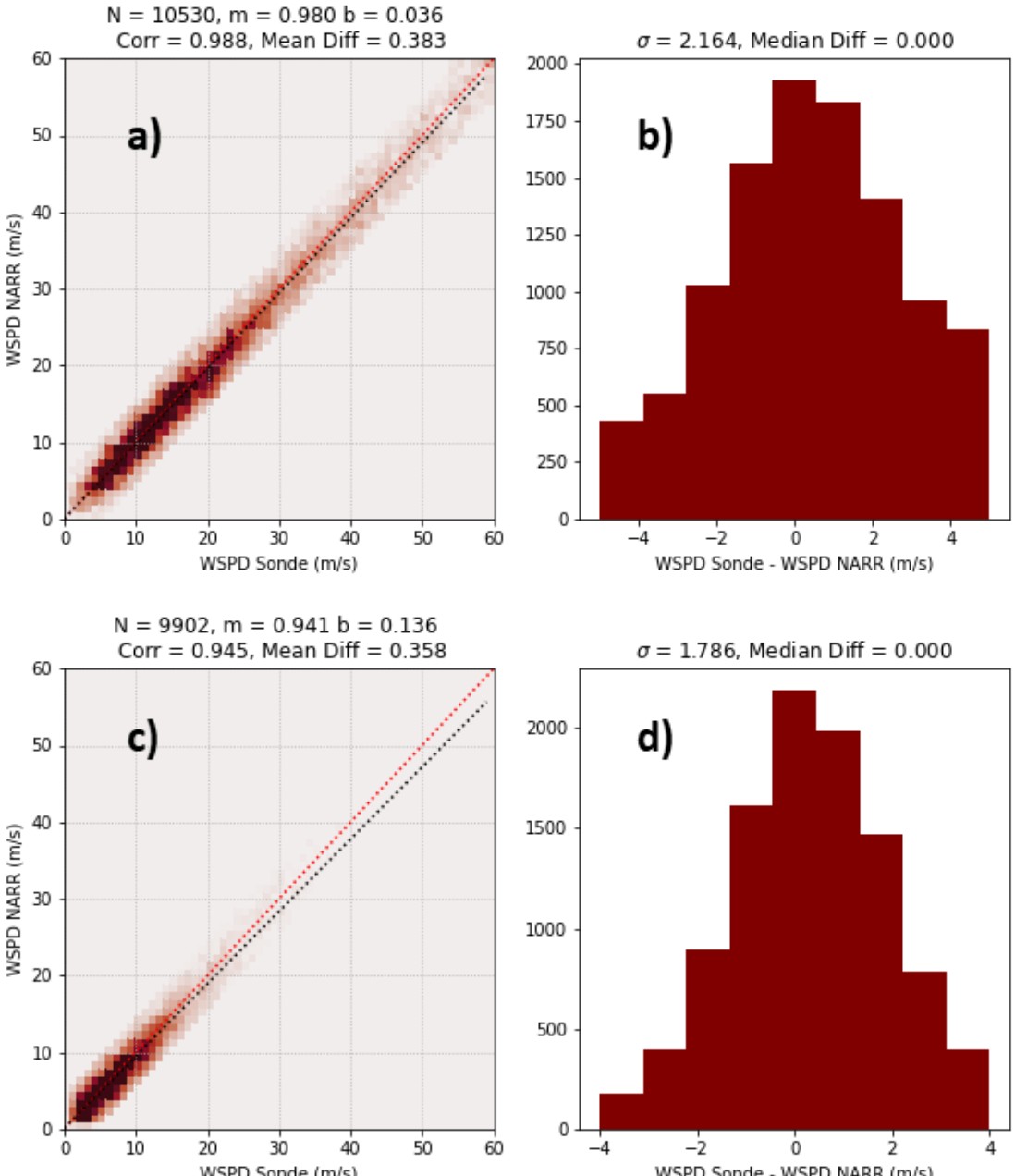

**Figure 2.** (**a**) scatter plot of the meteorological winter NARR wind speed values for NRMN vs. the OUN sounding wind speed values binned to $1 \text{ m s}^{-1}$. Black line is the least-squares regression, and the red dotted line is the 1-to-1 line. The title displays the slope (m), the intercept (b), the number of points (N), the Pearson correlation (Corr), and the mean difference (Mean Diff); (**b**) the two-dimensional histogram displays the differences in wind speeds between the sounding data and the NARR data for meteorological winter. The title displays the standard deviation ($\sigma$) and the median difference (Median Diff); (**c**) similar to figure (**a**) for the summer months; (**d**) similar to figure (**b**) for the summer months.

The Pearson correlations for the wind speeds were high (0.988 for winter and 0.945 for summer). Such strong correlations reveal that the NARR on average is a valid approximation of the sounding data, especially considering the sounding values were interpolated onto the NARR pressure levels. However, Figure 3c,f demonstrate the large variability in individual values, especially at the lowest NARR pressure levels. This could be due to the assumptions about turbulent stress the NARR uses in its models. The temperature deviations tend to overestimate the values at the lowest NARR pressure

levels and underestimate at the higher NARR pressure levels. This was also seen in the wind speed deviations, which agrees with earlier findings [13]. One potential reason the NARR underestimated the wind speeds is because the wind speed values are averaged over the spatial grid, which could average out local wind maxima. This is consistent with earlier analysis [13]. Overall, our validation analysis supported the notion that the NARR was a viable option for the climatological study.

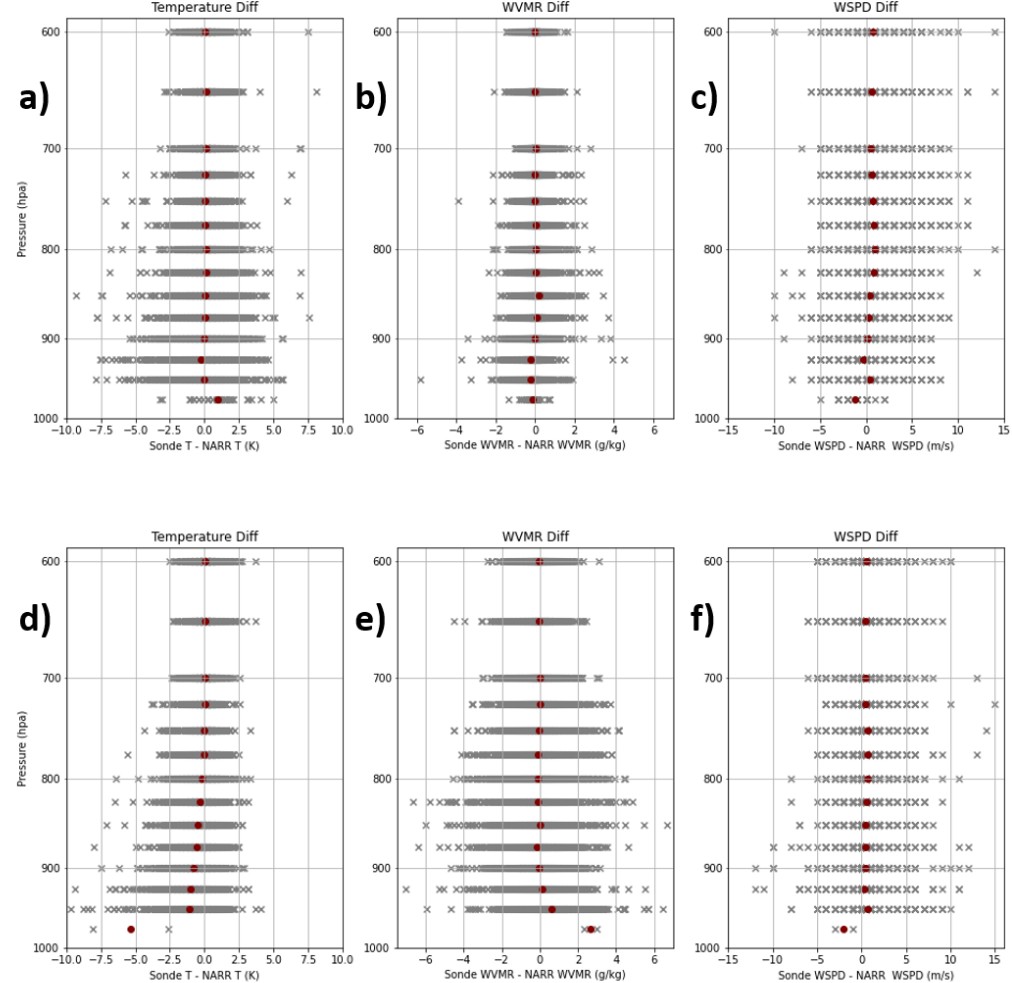

**Figure 3.** The differences in temperature (**a**), water vapor mixing ratio (**b**), and wind speed (**c**) for meteorological winter. Each x marks a difference, with the red symbol representing the mean. (**d**–**f**) are as (**a**–**c**), but for meteorological summer. differences are plotted with respect to atmospheric pressure on a logarithmic scale.

## 4. Results

### 4.1. Seasonal Climatologies

#### 4.1.1. Visibility

The fractions of successful hypothetical flights for each season with the visibility threshold as the only restriction are shown in Figure 4. Winter had the largest range of values from between 84% and 86% to greater than 98%. The lowest percentages were in the southeastern corner of the state. The fractions of successful hypothetical flights increased toward the northwest with the highest values in northwestern Oklahoma. Spring fractions of successful hypothetical flights ranged from 88% to between 96% and 98%. The lowest values were in the southeastern corner. Summer fractions of

successful hypothetical flights were lowest in the southeastern corner with values between 90% and 92%. The highest values, values greater than 98%, were found across most of the state. Autumn fractions of successful hypothetical flights were lowest in the southeastern corner of the state with values between 92% and 94%. Most fractions of successful hypothetical flights were between 96% and 98%. Isolated regions of values greater than 98% were found in northern and mid-eastern parts of the state.

Generally, the fractions of successful hypothetical flights were lower in the southeastern part of the state. Most of the state had fractions of successful hypothetical flights over 94% for all seasons, which suggests the visibility alone will not significantly prohibit the flight patterns of the WxUAS.

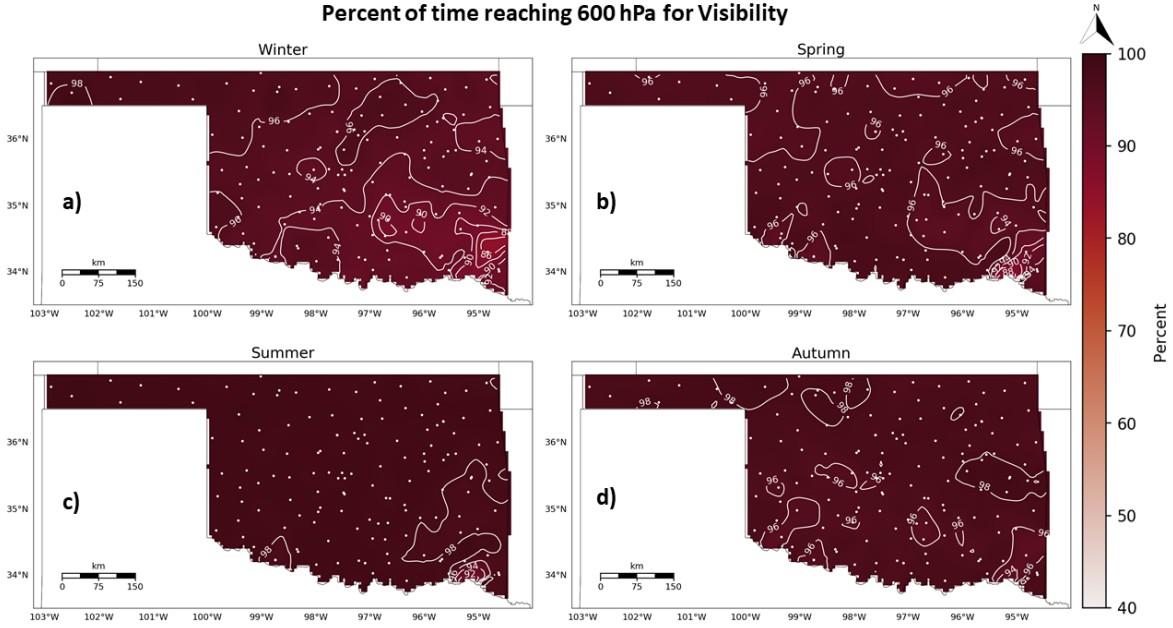

**Figure 4.** The fractions of successful hypothetical flights of UAS profiles given a minimum visibility of 4828 m in (**a**) winter, (**b**) spring, (**c**) summer, and (**d**) autumn. Each dot represents an Oklahoma Mesonet Site.

### 4.1.2. CBL Height

The fractions of successful hypothetical flights for each season with the CBL height threshold as the only restriction are shown in Figure 5. In the winter months, more than half of the state had a fraction of successful hypothetical flights of less than 76%. The lowest values were in the southeastern corner and were between 58% and 60%. The fractions of successful hypothetical flights increased toward the western part of the state with the highest values between 86% and 88% in northwestern Oklahoma. In the spring months, the lowest values were in the northeast between 62% and 64%. The fractions of successful hypothetical flights increased toward the southwest and then west. The highest values were in northwestern Oklahoma between 84% and 86%. More than half of the state had fractions of successful hypothetical flights below 74%. In the summer months, the lowest values of fractions of successful hypothetical flights between 54% and 56% were in the southeastern corner. There was a sharp gradient between 60% and 68% in the southeast. There were areas with fractions of successful hypothetical flights in the mid 60% range in the northeastern portion of the state. The highest values of success were in northwestern Oklahoma between 86% and 88%. Most of the state had fractions of successful hypothetical flights below 74%. In the autumn months, the lowest values, fractions of successful hypothetical flights between 62% and 64%, were in the southeastern corner. The highest values between 84% and 86% were in northwestern Oklahoma. The fractions of successful hypothetical

flights increased toward the northwest with most fractions of successful hypothetical flights lower than 72%.

The summer had the largest range of values. Most fractions of successful hypothetical flights were in the 70% range with no fractions of successful hypothetical flights at or above 90%, which suggests that the CBL height parameter could significantly impact the flight patterns of the WxUAS under current regulations.

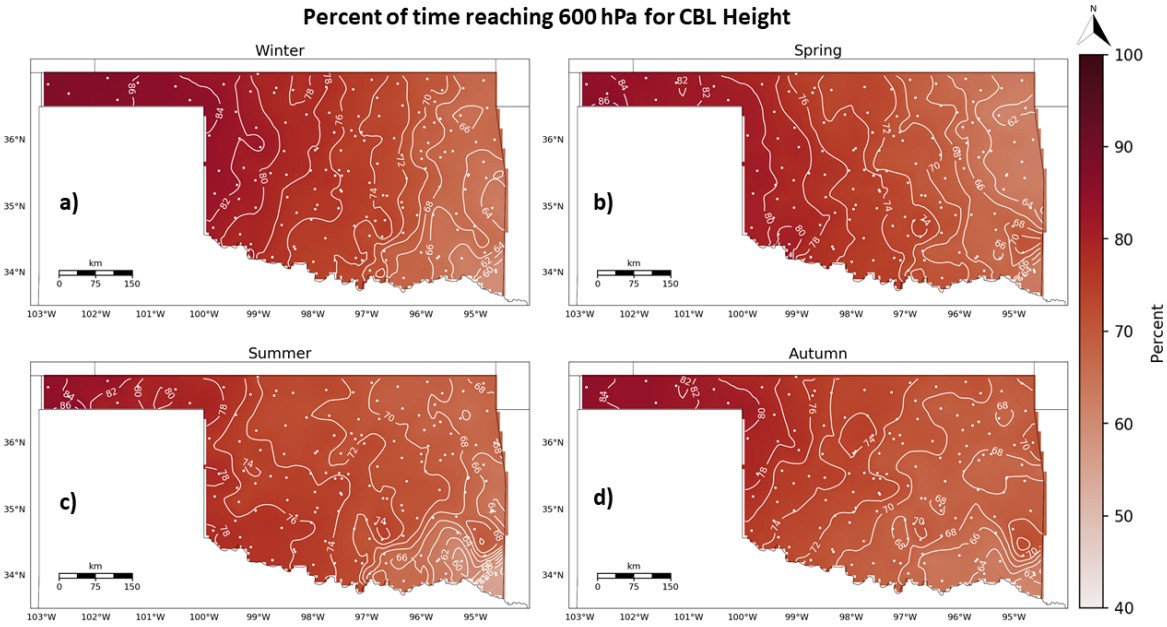

**Figure 5.** The fractions of successful hypothetical flights of UAS profiles given the CBL height restriction for (**a**) winter; (**b**) spring; (**c**) summer; and (**d**) autumn. Each dot represents an Oklahoma Mesonet Site.

### 4.1.3. Wind Speed

The fractions of successful hypothetical flights for each season with the wind speed threshold as the only restriction are shown in Figure 6. In the winter months, the values ranged from between 68% and 70% in the northeastern part of the state to between 88% and 90% in northwestern Oklahoma. The fractions of successful hypothetical flights increased toward the southwestern and western parts of the state with significant portion of the states fraction of successful hypothetical flights below 78%. In the spring months, the lowest values were in the northeast between 80% and 82% The fractions of successful hypothetical flights increased toward the western and southwestern parts of the state with a significant portion of the state lower than 86%. The highest fractions of successful hypothetical flights were in northwestern Oklahoma between 94% and 96%. In the summer months, no contour existed because all of the fractions of successful hypothetical flights were greater than 98%. In the autumn months, all of the fractions of successful hypothetical flights were higher than 92%. There were two contours on the map with increasing values from the northeastern to the southwestern parts of the state. The highest values were in the western part of the state and northwestern Oklahoma with values greater than 98%.

The winter fractions of successful hypothetical flights were the lowest overall with most values in the range of 70% with no fractions of successful hypothetical flights at or above 90%. Most spring fractions of successful hypothetical flights were in the 80 s range. These values suggest the wind tolerance threshold parameter could significantly impact the flight patterns of the WxUAS in the winter and the spring.

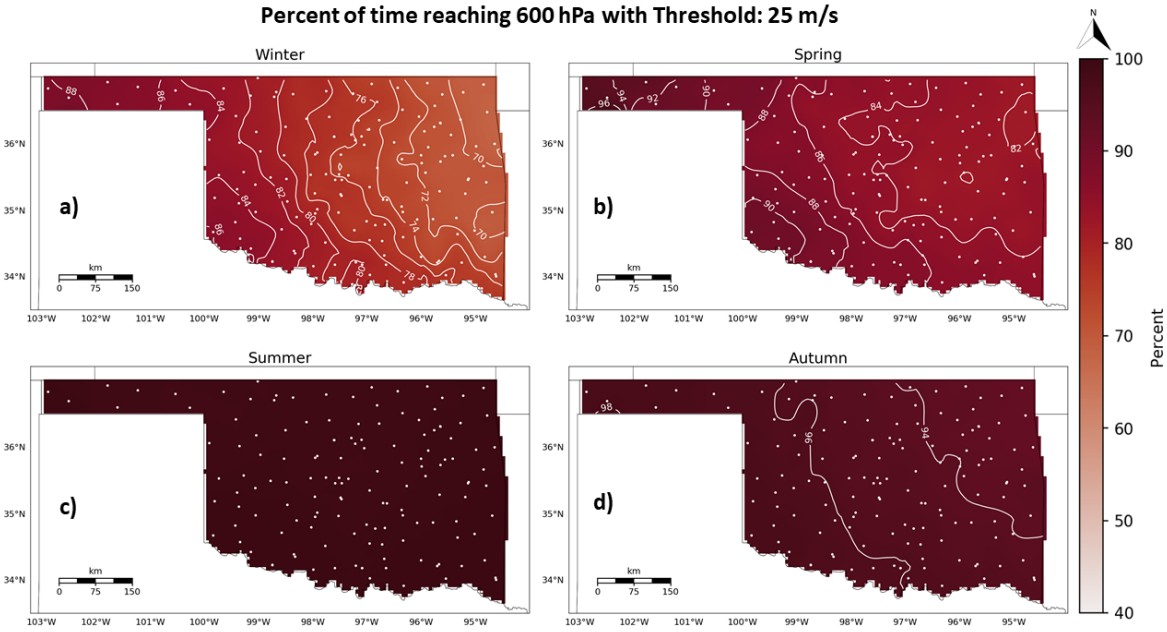

**Figure 6.** The fractions of successful hypothetical flights of UAS profiles given a maximum wind speed tolerance of 25 m s$^{-1}$ for (**a**) winter; (**b**) spring; (**c**) summer; and (**d**) autumn. Each dot represents an Oklahoma Mesonet Site.

### 4.1.4. Combined Climatologies

The fractions of successful hypothetical flights for each season with all restrictions applied are shown in Figure 7. Winter success values were lowest in the southeastern corner of the state between 42% and 44%. The fractions of successful hypothetical flights increased toward the west with the highest values between 76% and 78% in northwestern Oklahoma. Most of the fractions of successful hypothetical flights were below 58%. Spring fractions of successful hypothetical flights were lowest in the northeastern corner of the state between 50% and 52%. The fractions of successful hypothetical flights increased toward the southwest and then west with the highest fractions of successful hypothetical flights with values between 80% and 82% in northwestern Oklahoma. Most values were below 60%. Summer success values were lowest in the southeastern corner of the state between 54% and 56%. There was sharp gradient of fractions of successful hypothetical flights between 56% and 66% in the southeast. Values in the low 60% range were in the northeast as well. In general, the fractions of successful hypothetical flights increased toward the southwest and west. The highest values were in northwestern Oklahoma between 84% and 86%. Most fractions of successful hypothetical flights were below 72%. Autumn success values were lowest in the southeastern corner of the state between 56% and 58%. The fractions of successful hypothetical flights increased toward the northwest with the highest fractions of successful hypothetical flights between 82% and 84% in northwestern Oklahoma. Most of the values were below 66%.

The lowest fractions of successful hypothetical flights were in the southeastern corner except for in the spring where the lowest values were in the northeastern corner. All of the highest values were in northwestern Oklahoma with the fractions of successful hypothetical flights increasing toward the western part of the state. The values were lower than expected and this will be discussed further in the next section.

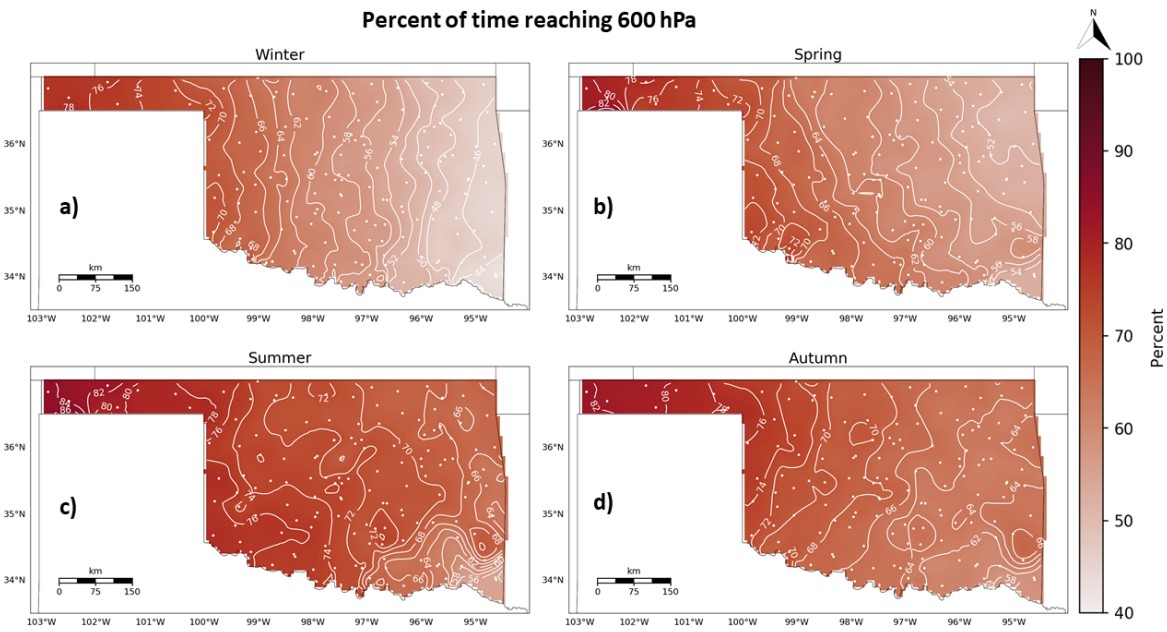

**Figure 7.** The fractions of successful hypothetical flights of UAS profiles for (**a**) winter; (**b**) spring; (**c**) summer; and (**d**) autumn. Each dot represents an Oklahoma Mesonet Site.

*4.2. Maximum Potential UAS Sampling Height AMSL Statistics*

The statistics of theoretical maximum flight altitude given wind speed tolerance, CBL height, and combined climatologies were calculated for the nine selected Mesonet sites. The Idabel, Oklahoma (IDAB) site has an elevation of 110 m above mean sea level and is located in the southeastern corner of the state where the fractions of successful hypothetical flights tended to be the lowest and the Kenton, Oklahoma, (KENT) site has an elevation of 1322 m above sea level and is located in the western part of northwestern Oklahoma where the fractions of successful hypothetical flights tended to be the highest. Therefore, we compared these two sites to discern how the limiting parameters affected the statistics of theoretical maximum flight altitude. Results are presented in Figures 8 and 9. Results for the other eight sites can be found in Figures S13–S32 in the supplemental document.

Box plots showing the maximum heights AMSL reached with only the wind speed threshold taken into consideration are shown in Figure 8a,b and box plots of the maximum heights AMSL reached considering only the CBL height threshold are shown in Figure 8c,d. For the wind speed box plots, the maximum height AMSL outliers reached below 1 km in every season except the summer for IDAB, and the lowest maximum height AMSL outliers were about 2 km for KENT. Note that outliers are defined here as points greater than $Q3 + 1.5IQR$ and less than $Q1 - 1.5IQR$, where $Q1$ and $Q3$ represent the first and third quartiles, and $IQR = Q3 - Q1$ is the interquartile range. Although there were more outliers for the IDAB site, the median values were roughly the same for both sites with values above 4 km. For the IDAB site, the interquartile ranges increased for the CBL height with the lower bound below 4 km for all seasons. However, the medians were still all above 4 km, which follows from the fraction of successful hypothetical flights percentage. The lower tails were all above 3 km or 700 hPa. For the KENT site, the CBL height interquartile ranges were much smaller than that of IDAB. All of the failures were shown as outliers and these values were all between 3.5 km and 4 km or between 650 and 600 hPa. The lack of outliers for the IDAB site compared to KENT when considering CBL height is likely related to a larger interquartile range of ASML maximum heights at IDAB thereby increasing the threshold to determine the presence of outliers.

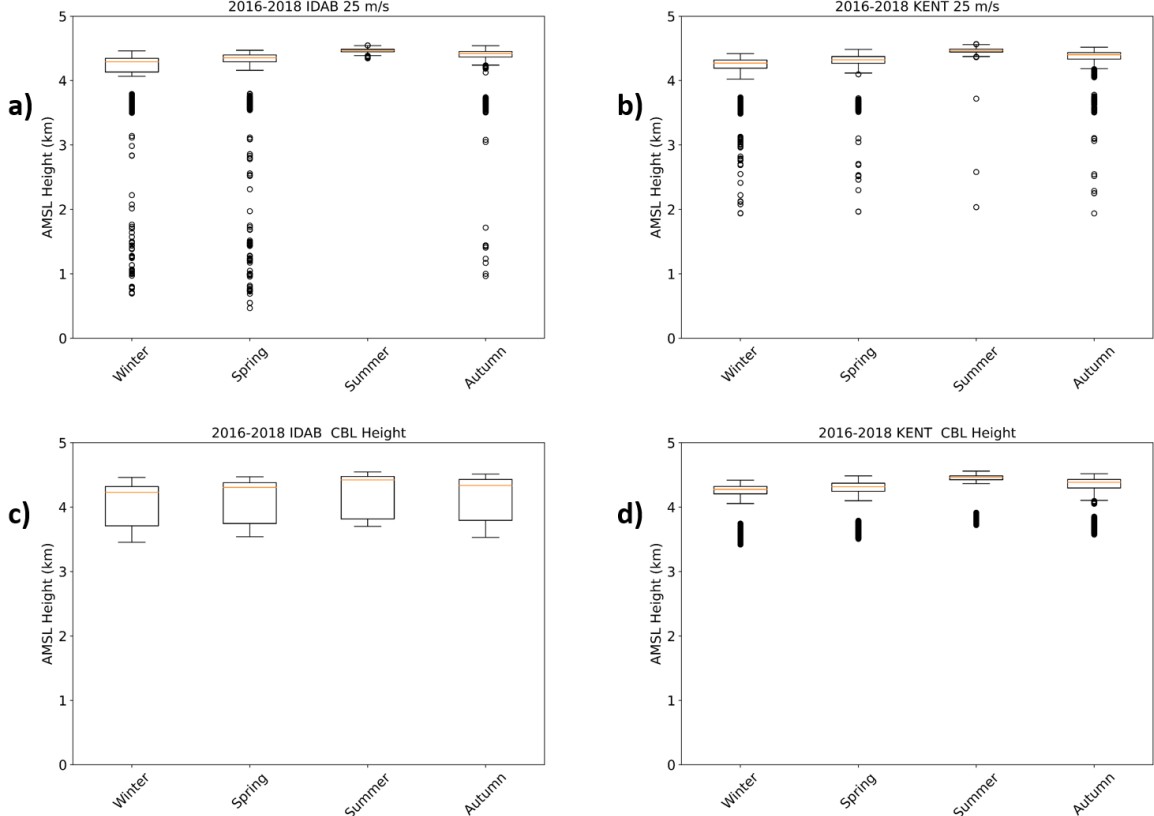

**Figure 8.** (**a**) the maximum height AMSL reached for the IDAB site given a maximum wind speed tolerance of 25 m s$^{-1}$. The circles are the outliers, and the yellow line is the median. All circles are open circles; (**b**) similar to figure (**a**), but for the KENT site. (**c**) the maximum height AMSL reached for the IDAB site given the CBL height restriction. The circles are the outliers, and the yellow line is the median. All circles are open circles; (**d**) similar to figure (**c**), but for the KENT site.

Next, we examine the box plots of the maximum heights AMSL reached at the IDAB and KENT sites with all thresholds imposed, including visibility. These are presented in Figure 9a,b. The circles at 0.11 and 1.32 km indicate the visibility less than 4828 m for the IDAB and KENT sites respectively, so the UAS would not be able to leave the ground. For the IDAB site, the median for winter dropped below 4 km, which again demonstrated a fraction of successful hypothetical flights of below 50%. The interquartile ranges for all seasons were about 1 km indicating a broader distribution of altitudes. For the KENT site, the interquartile ranges for all seasons were larger compared to the ranges for wind speed and CBL height alone. All medians were above 4 km.

The median maximum heights AMSL for the selected Mesonet sites are shown in Table 3. The median values were used because the geopotential heights; therefore, the AMSL heights, change for each NARR profile. The values were lowest in the winter and highest in the summer with sites in the western parts of the state achieving higher medians than the eastern parts of the state. Idabel had the lowest medians overall and Kenton had the highest medians overall. In the winter, the IDAB, MIAM, and PORT sites had values less than 4 km indicating fractions of successful hypothetical flights under 50%. These results are consistent with the fractions of successful hypothetical flights for the combined climatologies. The fractions of successful hypothetical flights for each site are shown is Table 4. These fractions of successful hypothetical flights were calculated using the method for combined climatology percentages for each Mesonet site.

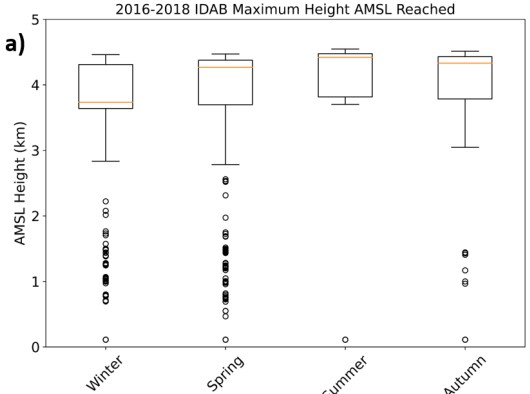
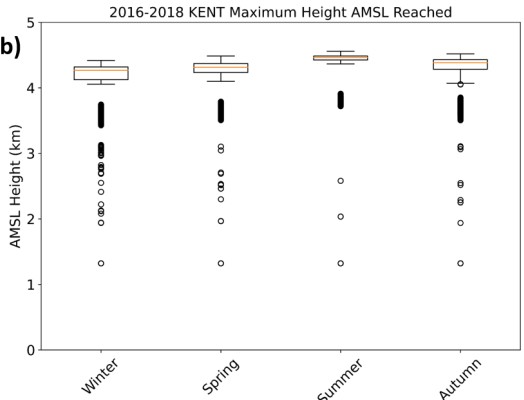

**Figure 9.** The maximum height AMSL reached for (**a**) the IDAB site and (**b**) the KENT site. The open circles are the outliers, and the yellow line is the median. All circles are open circles.

**Table 3.** The median maximum height AMSL reached for each season at the nine selected Oklahoma Mesonet sites.

| Site ID | Elevation (km) | Winter (km) | Spring (km) | Summer (km) | Autumn (km) |
|---|---|---|---|---|---|
| FITT | 0.35 | 4.22 | 4.30 | 4.46 | 4.36 |
| HOBA | 0.47 | 4.26 | 4.32 | 4.46 | 4.37 |
| IDAB | 0.11 | 3.73 | 4.27 | 4.42 | 4.33 |
| KENT | 1.32 | 4.27 | 4.31 | 4.46 | 4.38 |
| MEDF | 0.33 | 4.22 | 4.27 | 4.45 | 4.35 |
| MIAM | 0.25 | 3.71 | 4.22 | 4.44 | 4.35 |
| NRMN | 0.36 | 4.23 | 4.30 | 4.46 | 4.36 |
| PORT | 0.19 | 3.74 | 4.25 | 4.45 | 4.34 |
| PUTN | 0.59 | 4.25 | 4.31 | 4.46 | 4.37 |

**Table 4.** The fractions of successful hypothetical flights for each season at the nine selected Oklahoma Mesonet sites.

| Site ID | Elevation (km) | Winter (%) | Spring (%) | Summer (%) | Autumn (%) |
|---|---|---|---|---|---|
| FITT | 0.35 | 56.6 | 61.5 | 71.6 | 70.5 |
| HOBA | 0.47 | 66.7 | 68.9 | 75.8 | 70.0 |
| IDAB | 0.11 | 44.0 | 52.9 | 54.9 | 58.1 |
| KENT | 1.32 | 76.8 | 80.8 | 84.7 | 82.6 |
| MEDF | 0.33 | 61.8 | 61.3 | 72.2 | 69.1 |
| MIAM | 0.25 | 46.5 | 51.7 | 68.0 | 65.2 |
| NRMN | 0.36 | 58.5 | 62.0 | 73.7 | 68.0 |
| PORT | 0.19 | 47.8 | 53.0 | 68.7 | 62.6 |
| PUTN | 0.59 | 67.0 | 67.8 | 74.0 | 73.5 |

## 5. Discussion

Based on the results of the validation process, the data from the NARR model were used in seasonal climatological analysis. These values were calculated with the current restrictions imposed by the FAA, so if WxUAS flights were permitted to fly in clouds or the visibility requirement was lowered, the fractions of successful hypothetical flights would increase. The seasonal climatologies study revealed that the CBL height parameter imposed the largest restriction on successful flights. This is evident in the shape and location of the contours on the combined climatologies maps, and the fractions of successful hypothetical flights were lower for the CBL height parameter than for the wind speed and the visibility parameters. The amount of annual precipitation could contribute to this with most precipitous cloud base level heights under the desired 4 km AMSL (around 600 hPa). Annual precipitation is higher in the southeastern portion of the state and northwestern Oklahoma experiences

the least amount of annual rainfall on average [22]. Our findings of higher fractions of successful hypothetical flights in northwestern Oklahoma and lower fractions of successful hypothetical flights in the southeastern corner are consistent with the average rainfall rates. The wind speed parameter impacted the winter the most with percentages of achieving the target altitude of 600 hPa in the 70% range. This could be due to a number of factors including the location of Oklahoma and the larger temperature differentials across the state. Some of the largest temperature differentials within a diurnal cycle occur in the winter due to arctic cold fronts [23]. The visibility parameter contributed the least to failures overall, with almost all values over 90%. Visibility values were around what was expected since the Mesonet sites are part of a rural network [5]. The maximum height AMSL statistics indicated the wind speed restriction and the restriction did not individually alter the median by a significant amount. The interquartile ranges were larger for the IDAB site indicating that lower values between 3 and 4 km were more common. The mean and median maximum height AMSL reached for winter at IDAB accounting for all three restrictions was under 4 km indicating less than half of the flights were successful. Given the lowest values in the lower tail were above 3 km or 700 hPa, the fractions of successful hypothetical flights may not completely reflect the extent to which the WxUAS could fly but do give an estimate of the percentage of time complete profiles can be sampled. Because the fractions of successful hypothetical flights were predicated off of a 600 hPa theoretical upper bound, if the profile can be accomplished with lower maximum height AMSL, the fractions of successful hypothetical flights will increase.

For the combined climatologies, the lowest fractions of successful hypothetical flights were in the southeastern corner except for in the spring where the lowest values were in the northeastern corner. For every season, the highest values were in northwestern Oklahoma with the fractions of successful hypothetical flights increasing toward the western part of the state. This could be due in part to the topography of Oklahoma [22,24]. Further investigation needs to be completed to determine the amount to which topography would affect the flight patterns of the WxUAS.

## 6. Conclusions

Our goal was to use the NARR data to determine the climatologies of potential WxUAS flight locations. To do so, the NARR data accuracy had to be verified. In our comparisons with the OUN radiosonde data, the NARR on average agreed with the sounding data for temperature, water vapor mixing ratio, and wind speed for the meteorological seasons. Based on these findings, we determined that NARR data could be used to determine fractions of successful hypothetical flights of vertical profiles using limiting parameters of visibility, CBL height, and wind speed. We found that CBL height was the largest limiting factor, though the wind speed limited the fractions of successful hypothetical flights in the winter. The western part of the state, particularly northwestern Oklahoma has the highest fractions of successful hypothetical flights, and the southeastern corner performs the worst in the winter, summer, and autumn. Future work will study the potential effect of topology and additional variables, such as amount of rainfall and temperature, on fractions of successful hypothetical flights by region of the state. This paper establishes a framework to analyze the NARR data to determine potential WxUAS locations.

**Supplementary Materials:** The following are available online at http://www.mdpi.com/2072-4292/12/18/2947/s1, Figure S1: Scatter plot of the meteorological winter NARR temperature values for NRMN vs. the OUN sounding temperature and two-dimensional histogram of the differences in temperature between the sounding data and the NARR data for meteorological winter, Figure S2: Scatter plot of the meteorological spring NARR temperature values for NRMN vs. the OUN sounding temperature and two-dimensional histogram of the differences in temperature between the sounding data and the NARR data for meteorological spring, Figure S3: Scatter plot of the meteorological summer NARR temperature values for NRMN vs. the OUN sounding temperature binned and two-dimensional histogram of the differences in temperature between the sounding data and the NARR data for meteorological summer, Figure S4: Scatter plot of the meteorological autumn NARR temperature values for NRMN vs. the OUN sounding temperature and two-dimensional histogram displays the differences in temperature between the sounding data and the NARR data for meteorological autumn, Figure S5: Scatter plot of the meteorological winter NARR water vapor mixing ratio values for NRMN vs. the OUN sounding

water vapor mixing ratio and two-dimensional histogram of the differences in water vapor mixing ratio between the sounding data and the NARR data for meteorological winter, Figure S6: Scatter plot of the meteorological spring NARR water vapor mixing ratio values for NRMN vs. the OUN sounding water vapor mixing ratio and two-dimensional histogram of the differences in water vapor mixing ratio between the sounding data and the NARR data for meteorological spring, Figure S7: Scatter plot of the meteorological summer NARR water vapor mixing ratio values for NRMN vs. the OUN sounding water vapor mixing ratio and two-dimensional histogram of the differences in water vapor mixing ratio between the sounding data and the NARR data for meteorological summer, Figure S8: Scatter plot of the meteorological autumn NARR water vapor mixing ratio values for NRMN vs. the OUN sounding water vapor mixing ratio and two-dimensional histogram of the differences in water vapor mixing ratio between the sounding data and the NARR data for meteorological autumn, Figure S9: Scatter plot of the meteorological spring NARR wind speed values for NRMN vs. the OUN sounding wind speed values and two-dimensional histogram of the differences in wind speeds between the sounding data and the NARR data for meteorological spring, Figure S10: Scatter plot of the meteorological autumn NARR wind speed values for NRMN vs. the OUN sounding wind speed values and two-dimensional histogram of the differences in wind speeds between the sounding data and the NARR data for meteorological autumn, Figure S11: The differences in temperature, water vapor mixing ratio, and wind speed as a function of altitude for meteorological spring, Figure S12: The differences in temperature, water vapor mixing ratio, and wind speed as a function of altitude for meteorological autumn, Figure S13: 2016–2018 FITT 25 m/s, Figure S14: 2016–2018 FITT CBL Height, Figure S15: 2016–2018 FITT Maximum Height AMSL Reached, Figure S16: 2016–2018 HOBA 25 m/s, Figure S17: 2016–2018 HOBA CBL Height, Figure S18: 2016–2018 HOBA Maximum Height AMSL Reached, Figure S19: 2016–2018 MEDF 25 m/s, Figure S20: 2016–2018 MEDF CBL Height, Figure S21: 2016–2018 MEDF Maximum Height AMSL Reached, Figure S22: 2016–2018 MIAM 25 m/s, Figure S23: 2016–2018 MIAM CBL Height, Figure S24: 2016–2018 MIAM Maximum Height AMSL Reached, Figure S25: 2016–2018 NRMN 25 m/s, Figure S26: 2016–2018 NRMN CBL Height, Figure S27: 2016–2018 NRMN Maximum Height AMSL Reached, Figure S28: 2016–2018 PORT 25 m/s, Figure S29: 2016–2018 PORT CBL Height, Figure 30: 2016–2018 PORT Maximum Height AMSL Reached, Figure S31: 2016–2018 PUTN 25 m/s, Figure S32: 2016–2018 PUTN CBL Height, Figure S33: 2016–2018 PUTN Maximum Height AMSL Reached.

**Author Contributions:** Conceptualization, A.M.J., T.M.B., B.R.G., and P.B.C.; methodology, A.M.J., T.M.B., B.R.G., and P.B.C.; software, A.M.J. and T.M.B.; validation, A.M.J.; formal analysis, A.M.J.; investigation, A.M.J., T.M.B., B.R.G., and P.B.C.; data curation, A.M.J.; writing—original draft preparation, A.M.J.; writing—review and editing, T.M.B., B.R.G., and P.B.C.; visualization, A.M.J.; supervision, P.B.C.; project administration, P.B.C.; funding acquisition, P.B.C. All authors have read and agreed to the published version of the manuscript.

**Funding:** This work has been supported by the Vice President for Research and Partnerships (VPRP) at the University of Oklahoma (OU).

**Acknowledgments:** The authors would like to extend their gratitude to Elinor Martin at the University of Oklahoma for providing us access to the NARR data used in this study.

**Conflicts of Interest:** The authors declare no conflict of interest.

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
