# Peer review of "The Effect of Climatological Variables on Future UAS-Based Atmospheric Profiling in the Lower Atmosphere"

_remotesensing, doi:10.3390/rs12182947_

Round 1

Reviewer 1 Report

" The effect of Climatological Variables on UAS-Based Atmospheric Profiling in the Lower Atmosphere".

In the abstract you say cloud base is important. Why? Do you need to fly below cloud base only? Please explain in the abstract.

Please give a brief description of the CopterSonde. How is it unique.

Line 123-138, It is not clear to me, are you allowed to fly between clouds or are you strictly below cloud base. Please explain more clearly.

Line 157-160. The way it is written is disturbing. Did you do all seasons? If so, why put it in a separate document. You can just say you did it all and only show the necessary parts.

Line 212. The way it is written "success rates" is a bit disturbing. Low visibility is not a failure, just a reality. Unless you want to convey the concept that you were not able to fudge the data to get the result you wanted. Maybe say ... " The fraction of cases which meet the visibility standard ... "

Line 226 Again using the term "success rates". Find another way to say it. Remove the term "success rate" from the paper.

Line 260 (and elsewhere) Change term "success rates".

Figure 8. In the figure caption please explain what the boxes and dots are.

Line 315-316. Please explain again here your definitiion of success and how cloud base affected the "success" rate. Again I suggest using another term instead of success.

Reviewer 2 Report

This is a sound paper on a relevant topic which merits publication.

The only problem I have with the paper is that, while it is all very well to say that a robotic aircraft can usually attain its target elevation, it is surely when it would be most needed, i.e. in foul weather, that it would fail.  The authors, after all, start the paper by pointing out the financial losses arising from  climate related disasters.  Of course, this makes the analysis in the paper all the more valuable when assessing the value of robotic aircraft.  Perhaps the authors could add a sentence or two in the discussion regarding this point.    

I'm not quite sure of the relevance of the 25 m/s limit.  Clearly it will be difficult to launch or retrieve the aircraft if the surface wind speed exceeds 50 kt, but once aloft it is surely only local turbulence that matters?  Is the risk that the aircraft would be lost downwind?

I presume that the WxUAS allows for any lag of the temperature and humidity sensors?  If the aircraft is climbing or descending rapidly this can be an issue.

Figures 2(b) and 2(d) could do with a legend on the ordinates.  'Numbers of observations'.

The outliers on the box plots, Figures 8&9, haven't come out very well.  Until I expanded them 200% I got the impression that there were solid circles and open circles and wondered what the significance was.  It's interesting that some plots have no outliers, i.e. that all measurements were within 2 s.d. of the mean.  Perhaps a few comments about the distribution sometimes having no tails might be appropriate.

There are a few minor typos, but these should drop out on sub-editing.

Reviewer 3 Report

Review of “The Effect of Climatological Variables on UAS-Based Atmospheric Profiling in the Lower Atmosphere” by Ariel M. Jacobs et al.

In this manuscript, a regional climate model is used to estimate the maximum altitude to which an unmanned aerial vehicle, equipped with meteorological sensors, could potentially fly, based on three years’ worth of data. Three factors are considered both individually and in combination: visibility, cloud base height and maximum wind speed. Results are presented on a seasonal basis. The ability of an aircraft to potentially reach a pressure of 600 hPa was deemed successful, in this study. The maximum possible sampling altitude was then analysed (in metres) at nine individual locations across Oklahoma, USA. A crucial benefit to strengthen the outcomes of this work, was that the authors tested the model, by comparing the model output (at fixed pressure levels) with measurements acquired from daily radiosonde launches.

This work serves as a useful example for future researchers wising to use unmanned aerial vehicle sampling in meteorological sampling. Good planning will be essential in this growing field, whist ensuring adequate safety and compliance with airspace regulations. Yet there are a few key issues with the presentation of results and some terminology. The title of the manuscript is also misleading. I strongly recommend that this work be published once the following issues have been addressed. This is a useful contribution to the field scientific unmanned aerial vehicle research and planning.

General comments

Introduction

  • This section is well written and suitably motivates the rest of the manuscript.
  • I am concerned with use of the term “Atmospheric Boundary Layer (ABL)”, both here and throughout the manuscript. I understand this to be the layer of the atmosphere nearest to the surface. This layer can vary in height over time. It should be clearly defined here and then used consistently.

Data and Methods

  • I am slightly confused with the way this section is organised. A short paragraph at the start may help guide the reader towards its contents.
  • It took me many hours to realise that this is a study looking at the potential use of UAS sampling, not actual UAS sampling, as implied in the title. It must be far clearer that this study looks at the potential use of UASs to acquire meteorological measurements. The word “potential” must be added when referring to UAS flights, throughout the manuscript.
  • I found definitions of heights very confusing. It’s not clear to me which heights are being used where, for example, what is a “maximum value height”?
  • The authors must clearly define what a “success rate” is. It is a heavily used term that is used quite casually. I assume a “success rate” to be the probability that a UAS flight could potentially be conducted at a certain season in a certain place. Consider replacing “success rate” with a different term such as “successful sampling possibility” or something better, throughout the manuscript.

Validation

  • This is generally a useful section to support the climatological analysis. This test provides great strength to the conclusions of the manuscript.
  • In this test, the model appears to output data at altitude levels in units of pressure whereas the subsequent climatological analysis plots metric altitude (compare figure 3 to figure 8). The authors either need to use a consistent method of presenting altitude throughout the manuscript (I would suggest sticking to hectopascals) or they must make it very clear how they convert from hectopascals into metres. Do the authors consider surface pressure changes in such a conversion?

Results

  • Maximum heights at two different locations is presented. I think it may be better to present these results in hectopascals, rather than in metres. As the model provides output at discrete pressure levels, it may be useful to provide the mode of maximum pressure levels for each location for each season. I’m not too sure about providing a the median, as (in theory) there should not be a continuous distribution in maximum height. Maximum heights should be at discrete pressure levels.
  • The term “success rates” must be clearly defined and used consistently.
  • The authors must be careful using the words “height”, “elevation” and “altitude”. I would suggest defining altitude above mean sea level (AMSL) and elevation (height above the ground). The authors should refrain from use of the word “height” too causally. Correct use of “AMSL” and “elevation” should instead be applied.

Discussion

  • This is a good, concise section.

Conclusion

  • This section suitably summarises the manuscript.

Specific comments

Page 1; Title: “The Effect of Climatological Variables on UAS-Based Atmospheric Profiling in the Lower Atmosphere”

  • I do not like this title as it implies that actual UAS sampling was conducted in this work. It is quite misleading. I would suggest inserting the word "potential" or "future" before "UAS" in the title.

Page 1; Line 3: “weather observing unmanned aircraft systems (WxUAS) to perform the”

  • Replace with “weather observing unmanned aerial systems (WxUAS) in future, to perform”.
  • WxUAS is a strange acronym. What does the “x” stand for?

Page 1; Line 4: “ascending into the lower ABL”

  • Consider rephrasing this as it is slightly vague. I understand the ABL to be the lowest part of the atmosphere. So the lowest part of the ABL is anywhere near 0 m in altitude. Ascending into the lowest part of the ABL could mean anything. Maybe replace this with “ascending to the top of the ABL”, provide a rough number (for example, “to approximately 100 m in altitude”) or remove this entirely.

Page 1; Line 10: “success rates”

  • Please define this.

Page 1; Line 10: “reach a pressure level”

  • Replace with “reach an altitude corresponding to a pressure level”.

Page 1; Line 13: “success rates”

  • Please be more specific with regards to what a “success rate” means. Do you mean successful correspondence between the model and the observations? How do you determine if a flight would be “successful”?

Page 1; Line 21: “year,”

  • Replace with “year;” or “year.” to start a new sentence.

Page 1; Line 22: “meteorological measurements are vital for the decision-making process in energy security, food production, public health and safety, transportation, and water resources”

  • This sentence does not appear to be grammatically correct. The “decision making process in energy security” doesn't really mean anything. Do you mean the decision making process in planning for changes in energy security, ?

Page 1; Line 24: “27 states”

  • Replace with “27 USA states” if that is what you mean. Other countries have states too.

Page 2; Line 30: “temperature”

  • Replace with “temperature,” (inserting a comma), to keep Oxford comma use consistent throughout the manuscript.

Page 2; Line 40: “unattended WxUAS”

  • Please elaborate here. Do you mean the UASs would fly automatically and unattended? This would be interesting.

Page 2; Line 41: “the numerical weather prediction models”

  • Replace with “numerical weather prediction models”, removing the definite article.

Page 2; Line 50: “the WxUAS can fly”

  • Replace with “the WxWAS would be able to fly”, to make it clear that you are referring to a potential UAS in future.

Page 2; Line 54: “used of analysis”

  • Replace with “used for analysis”.

Page 2; Line 65: “Significant advantages of the NARR model include spatial and temporal resolution of 32-km”

  • How can both the spatial and temporal resolution be 32 km? I'm slightly confused.

Page 2; Line 72: “We then used the NARR data from the point closest to the geographical location of the mesonet site.”

  • If possible, provide minimum distance, maximum distance, mean and the standard deviation between the location of mesonet sites and corresponding NARR data point centres, in metres.

Page 3; Line 78: “closest proximity to Norman”

  • Exactly how far from the centre of the grid is Norman, in metres?
  • Please provide coordinates for Norman in brackets?

Page 3; Line 102: “Seasonal Climatologies”

  • Why study climate on a seasonal basis? Maybe add a short sentence somewhere motivating this seasonal study. You may wish to add something in the introduction, to further motivate this seasonal approach.

Page 3; Line 103: “success rates of the WxUAS”

  • Replace with “success rates of potential WxUAS sampling”?

Page 3; Line 110: “in Class C, D, E, and G airspace”

  • Either remove this or explain what these classes mean in the USA.

Page 4; Table 1: “failure”

  • Either replace this negative word or explain what you mean by “failure”.

Page 4; Line 124: “the flight does not occur”

  • If the UAS does not take off, the flight is obviously unsuccessful. Possibly replace with “a UAS flight is not permitted”.

Page 4; Line 126: “found the NARR estimates that cloud base”

  • Replace with “found that the NARR estimates cloud base”.

Page 4; Line 131: “If 152.4 m below the cloud base level height corresponded to a pressure greater than 600 hPa and a combined low cloud and medium cloud percentage greater than 12% the flight was considered unsuccessful and the maximum height value was set at the height corresponding to the lowest successful pressure level.”

  • Please consider inserting some commas or breaking this sentence down as it is very difficult to follow.
  • What does “the flight was considered unsuccessful” mean, if no UAS flights have yet taken place?
  • Maybe a cartoon diagram would help to clarify definitions of heights.

Page 4; Line 136: “reached above 25 m s-1 during the”

  • Replace with “exceeded 25 m s-1 during a”.

Page 4; Line 137: “at the height of the at”

  • Replace with “to”.

Page 4; Line 130: “Selected Sites”

  • Why are you looking at the maximum potential UAS heights at these 9 sites? Was there anything special about these sites or were they chosen at random?
  • Consider adding a short sentence to motivate this work. I think you are looking at specific mesonet locations for UAS sampling to complement stationary sampling. Nevertheless, it should be explicitly stated somewhere, to avoid confusion.

Page 4; Line 141: “reached by the UAS”

  • Replace with “could be reached by the UAS”.

Page 4; Line 144: “maximum height statistics”

  • Replace with “maximum potential UAS sampling height statistics” or something similar.

Page 4; Line 146: “combined conditions calculations”

  • What is a “combined conditions calculation”?

Page 5; Line 166: “Over 90% of the points used were within two standard deviations.”

  • This is interesting. 2 standard deviations should capture 95% of data points but you only use 90%. I'm intrigued regarding what happened to the other 5%.

Page 5; Line 168: “winter and summer”

  • Please provide precise dates of winter and summer somewhere in the manuscript. “Meteorological winter” varied from country to country in time and duration.

Page 5; Paragraph commencing on Line 161

  • You could consider giving all of these values in a table. It is slightly messy in the text.

Page 5; Line 164: “9902 data points”

  • Out of how many? Please specify.

Page 5; Line 165: “As shown in (Fig. 2c), 9902 data points corresponding to NARR wind speeds vs sounding wind speeds were within two standard deviations.”

  • You need to specify that you are talking about summer here. Otherwise it looks like you are repeating the same sentence twice. I thought you were still discussing winter.
  • I do not see 2 standard deviations in fig 2c. Maybe you mean fig 2d.

Page 6; Line 173: “are shown in Fig. 3”

  • Replace with “are shown in Fig. 3 as a function of height.”. This is a key difference between fig 2 and fig 3.

Page 6; Line 175: “mean differences winter temperature differences”

  • Replace with “mean winter temperature differences”.

Page 6; Line 176: “the mean”

  • Replace with “the mean difference”.

Page 7; Line 188: “The summer temperature plot, Fig. 3d, displayed the mean differences increased between -1 and 0 K up until 700 hPa.”

  • There may be a grammatical error somewhere here. Maybe replace with "The summer temperature plot, Fig. 3d, shows that the mean temperature difference increases from -1 K up to 0 K with increasing height (up to 700 hPa)."

Page 7; Figure 3

  • This is a good plot but it seems like some of the points are slightly cut off to the edges of the range. Would you be able to extend the x-axis of the plots please so that the full range can clearly be seen?

Page 7; Line 198: “The Pearson correlations for the wind speeds were high (0.988 for winter and 0.945 for summer).”

  • You already state this in chapter of section 3. I would suggest only stating it here and removing it from the earlier discussion (unless the earlier values are somehow different from these values).

Page 7: Line 202: “The temperature and water vapor mixing ratio deviations tend to overestimate the values in the lower ABL and underestimate in the upper ABL.”

  • It is slightly confusing if you refer to “lower ABL” and “upper ABL” without specifying what pressure you consider these regions to be. From my understanding the ABL can be identified on a case-by-case basis based on individual soundings and changes in surface pressure. The ABL height changes regularly. Either clearly define what you mean the ABL to be or omit any reference to the ABL from the manuscript.
  • Furthermore, I do not see water vapour estimates evolving from overestimation to underestimation with altitude, in fig 3.

Page 8: Line 212: “success rates”

  • Please define what a success rate is. Is it the chance of a successful UAS flight? This is not defined anywhere.

Page 8: Line 214: “The values increased”

  • What values increased?

Page 8: Figure 4

  • 4, Fig. 5 and Fig. 6 use the identical colour scales which is helpful. However, maybe an additional colour can carefully be used on the same scale, as it is very difficult to identify small changes in the success rate from this figure.
  • I would also suggest changing the colours of the contours and labels. It is difficult to see the white lines and text.
  • How were the contours generated? Was some sort of interpolation used? Please describe this either here or in the main text.

Page 10: Line 273: “The lows”

  • The “low” what?

Page 11: Line 277: “Maximum Height Statistics”

  • Why don't you plot visibility separately? There may be a good reason, that can be summarised in a sentence here.

Page 11: Line 278: “maximum-height-reached statistics”

  • This is a confusing term.

Page 11: Line 286: “with the wind speed threshold”

  • Replace with “with only the wind speed threshold”.

Page 11: Line 287: “considering the CBL height threshold”

  • Replace with “considering only the CBL height threshold”.

Page 12: Figure 8

  • The NARR model provides output at fixed pressure levels. Why is there such as continuous range in heights? I would expect the outliers to be clustered. The same goes for fig. 9 and fig. 10.

Page 13: Table 3

  • This is slightly strange. Success rates are determined by surpassing a 600 hPa pressure threshold whereas Table 2 presents median heights in km. Why not present the values in table 2 in hPa? How did you calculate the success rates in Table 3? Do you assume 600 hPa to be at a fixed altitude in metres?

Page 13: Line 314: “WxUAS”

  • Replace with “WxUAS flights”

Page 13: Line 319: “under the desired 4 km”

  • I thought 600 hPa was desired. Where did 4 km come from? The pressure choice should be consistent throughout the manuscript.

Page 13: Line 327: “overall almost all values over 90%”

  • Replace with “overall, with almost all success rates over 90%”.

Page 14: Line 349: “success rates”

  • Give a very short definition of success rates here.

Page 14: Line 363: “The potential effect of topology and other reasons, such as amount of rainfall and temperature, certain portions of the state perform better needs to be studied further.”

  • This sentence is difficult to follow.

Reviewer 4 Report

Major comments

This paper fits into the field of surface meteorology, so I think it would be better suited to a journal such as Atmosphere.

Although this paper carefully and compactly summarizes the material, it is structured rather like a Master's or Doctoral thesis. Therefore, it is necessary to revise the structure in keeping with the style of an academic paper. Moreover, the paper's structure is too compact, hence, there is a lack of information overall, and especially regarding the methods. If the aim is to eliminate the discrepancy between the surface observed data and the modeling data, it will be necessary to consider them from their respective viewpoints, so the explanation will inevitably increase. Being able to show a concrete solution would help in dealing with the observed values and modeling data, and this study will provide an approximate measure of the number of observations required to improve the modeling data.

It is certain that the study of the "lower atmospheric boundary layer" is very important for meteorological studies. There is no doubt that some gaps will occur due to different observation methods and altitudes (km scale). Resolving these gaps is one of the motivations for this research. However, it is a little unclear what relevance the gaps have for this paper, and it lacks concreteness because this is not explained. As a result, the significance of the paper is less clear, and clarifying this will improve the impression it gives.

This journal is an international open access journal, so it will have a diverse readership (not only American people). “Seasons” have different periods and characteristics depending on the latitude and environment. Therefore, you should explain the details of your study site, and the methods or reasoning, in the “data and method” section. Further, it is not clear why you selected these “nine climate divisions” in this study. You should also explain the reasons in Section 2.4. Moreover, it is necessary to explain the structure of the rawinsonde and the observation system used in this study, using information from the references in Section 2.1.2. The references alone are not enough for the reader to understand this. Please include not only the pressures but also the flight altitudes of rawinsonde in the method. Details of the rawinsonde flight method (observation method) such as altitude, flight time, the sensor's name and its accuracy, etc. should be included. Meteorological data at the time of observation should also be included (satellite data may be used). There may be non-negligible meteorological conditions during the cited observation period. It may be that these unusual weather conditions were excluded from the analysis, but it is difficult to read from the method. It may be that these unusual weather conditions were excluded from the analysis, but it is difficult to read from the method.

Again, what kind of statistical analysis method was used, how the seasons were defined, and how the data were selected should be described in more detail in Section 2.3. If vertical profiles are important, give information about the observational altitude of the rawinsonde. Include the calculation methods, such as the formula, and add the data set used in the calculation as an appendix.

Data from rawinsonde observations seems to be used. Is this what the authors observed, or is it only archive data? The writing will change accordingly. If rawinsonde observations were taken, the system and methods need to be explained. This would add a strong element of originality to this study (Section 2.1.2).

The seasonal variations and the success rates described in the “Results” includes interpretations. The results should be written more objectively, and the discussion of the seasonal variations moved to the "Discussion"; or delete the “Discussion” and change the “Results” to "Results and Discussion". Section 4.2 should include information about the observational method (rawinsonde?) used in the research, citation information (for archive data), and statistical calculations performed. It does not make sense as currently written.

Do the distribution and density of Mesonet sites, and the data density, affect the success rate? Despite the extensive appendix, this is a big question, as readers don't know the details of the data selected and analyzed (in the Results and Discussion sections).

For now, the “Discussion” is not very different from the “Conclusion”, so please make the discussion more detailed. Please cite the data you got and discuss in detail. And it is necessary to write how the results and discussions is useful for future data analysis and rawinsonde observation. Finally, the conclusion and abstract should be rewritten based on the above revisions.

Minor comments

L21: Please write a few sentences citing the previous research on the global situation such as "How the global climate is changing" or "what are the response measures" in the introduction. For example, the information given in Chilson et al. 2019 (you cited). Please recheck this.

L25: Add a description of the Mesonet, because other countries also have observation systems named "Mesonet".

L53-56: As this paper is not a thesis, this sentence is unnecessary.

L275-276: Please remove the last sentence, which is unnecessary.

L355-356: Please remove the last sentence, which is unnecessary.

Figures: Size of Figure 3, 8 and 9 is little small. Please resize it.

Figure: It will be easier to understand if you have a flow chart of your observations and data analysis.

Reference

Alghamdi, A.S. 2020. Evaluation of Four Reanalysis Datasets against Radiosonde over Southwest Asia. Atmosphere, 11(4), 402.

Challa V, Indracanti J, Rabarison M, Patrick C, Baham J, Young J,Hughes R, Hardy M, Swanier S, Yerramilli A (2009) A simulation study of mesoscale coastal circulations in Mississippi Gulf coast. Atmos Res., 91, 9–25.

Chiba, T., Haga, Y., Inoue, M., Kiguchi, O., Nagayoshi, T and Madokoro, H. 2019. Measuring regional atmospheric CO2 concentrations in the lower troposphere with a non-dispersive infrared analyzer mounted on a UAV, Ogata Village, Akita, Japan. Atmospere, 10(9), 487.

Desai, A.R., Davis, K.J., Senff, C.J., Ismail, S., Browell, E.V., Stauffer, D.R. and Reen, B.P. 2006. A case study on the effects of heterogeneous soil moisture on mesoscale boundary-layer structure in the southern Great Plains, U.S.A. Part I: Simple prognostic model. Bound.-Layer Meteor., 119, 195–228.

Golkar F., Al-Wardy, M., Saffari S.F., Al-Aufi, K. and Al-Rawas, G. 2020. Using OCO–2 Satellite Data for Investigating the Variability of Atmospheric CO2 Concentration in Relationship with Precipitation, Relative Humidity, and Vegetation over Oman. Water, 12(1), 101.

Kim, M.-S. and Kwon, B.H. 2019. Estimation of Sensible Heat Flux and Atmospheric Boundary Layer Height Using an Unmanned Aerial Vehicle. Atmosphere, 10(7), 363.

Schwartz, B.E. and Charles A. Doswell. C.A. 1991. North American Rawinsonde Observations: Problems, Concerns, and a Call to Action. Bull. Amer. Meteor. Soc. 72(12), 1885–1896.

Song H.-S., Kim, S., Lee, H. and Kim, K.-H. 2020. Climatology of Tropospheric Relative Humidity over the Korean Peninsula from Radiosonde and ECMWF Reanalysis. Atmospere, 11(7), 704.

Sun, Q., Vihma, T., Jonassen, M.O. and Zhang, Z. 2020. Impact of Assimilation of Radiosonde and UAV Observations from the Southern Ocean in the Polar WRF Model. ADVANCES IN ATMOSPHERIC SCIENCES,37,441–454.

Weber, B.L. and Wuertz, D.B. 1990. Comparison of Rawinsonde and Wind Profiler Radar Measurements. J. Atmos. Oceanic Technol. 7(1), 157–174.

Round 2

Reviewer 4 Report

The manuscript became more understandable by revisions. However, I would like you to keep a revision history in the revised manuscript by the function of Word, so please be careful next time (not this submission).

I don't think it is necessary to explain the structure of the paper in introduction section or give instructions on sections. If there are many directives, it is difficult to focus because the sentences have to be moved forward and backward when reading. Therefore, even if the amount of text increases, I think it should be shown concretely. Because it is easy for the reader to understand. However, I found that the structures that the authors claimed was a little strange, though there were some. The structure does not matter any more, but I would like you to reconsider it in the future.

In each section especially Methods, a lot of explanations were added, so that the assumptions, observation methods, and the ones used in the calculation became easy to understand.